# The Role of Pharmaceutical Scientists in the Formation of a Healthy Lifestyle as a Value Orientation

**DOI:** 10.3390/pharmacy10010020

**Published:** 2022-01-26

**Authors:** Larisa Galiy, Tetiana Lutaieva, Larysa Lenchyk, Oleksandr Surikov, Svitlana Moroz

**Affiliations:** Institute for Advanced Training of Pharmacy Specialists (IATPS),National University of Pharmacy, 17 Zakhysnykiv Ukrainy Sq., 61001 Kharkiv, Ukraine; larisa_galiy@ukr.net (L.G.); larysa.lenchyk@nuph.edu.ua (L.L.); aasurikov@gmail.com (O.S.); m.sg@ukr.net (S.M.)

**Keywords:** pharmaceutical scientists, healthy lifestyle, public and educational activities, health, health-saving educational environment, health-promoting educational technologies, value orientations, pharmaceutical education, Imperial Kharkov University, pharmaceutical interns, orientation to a healthy lifestyle by representatives of the institution of higher pharmaceutical education

## Abstract

This article focuses on the role pharmaceutical scientists play in achieving social well-being, in particular in the formation of a healthy lifestyle as a value orientation among students and the public in historical retrospect. The following set of research methods was used: a general scientific method, and a historical (chronological, logico-historical, retrospective), personalized, empirical method (including the questioning of pharmaceutical interns and the analytical processing of questionnaires). The territorial boundaries of the study cover Slobozhanshchyna, a historical and geographical region centered in Kharkiv. The educational activities of Slobozhanshchyna’s pharmaceutical scientists during the 19th and early-20th centuries, as well as in the modern period, are presented. Attention is drawn to the fact that the role of pharmaceutical scientists in forming the environment for the development of a harmonious, spiritually and physically developed personality in historical retrospect is important. It is noted that the founders of pharmaceutical education in Ukraine initiated educational activity as a means for promoting the formation of a healthy lifestyle in the educational space and outside of the Imperial Kharkov University. The changes in the public and educational activities recommended for forming a healthy lifestyle among Slobozhanshchyna’s medical scientists and naturalists during the Imperial era have been generalized. The analysis of the research sources and questionnaires of pharmaceutical interns allowed us to assert that, in modern times, Slobozhanshchyna’s pharmaceutical scientists are trying to organize socially useful activities, taking into account the best achievements of the founders of pharmaceutical education in Ukraine. The “orientation to a healthy lifestyle of representatives of the institution of higher pharmaceutical education” has been proposed. This includes the adoption of a set of material, social, and spiritual benefits and ideals considered to be objects of purpose in the pursuit of a healthy lifestyle by employees and students (i.e., part of the implementation of the mission of the University), and using these as tools to meet the needs of both university representatives and ordinary citizens. This article contains a number of factors that mediated the formation of a healthy lifestyle as a value orientation of an individual, including: (a) the circumstances of the individual’s life (the conditions of education of the individual); (b) the individual’s consciousness reflecting the general and specific conditions of their personality formation; (c) the motivation of the real behavior mediated by the subject’s activities. The presence of these factors was identified as typical for the health-saving educational environment. This article provides recommendations regarding the following measures for the organization and effective promotion of a healthy lifestyle by modern pharmaceutical scientists: provide an opportunity to study the peculiarities of the organization of educational activities by scientists of the past to students and specialists in the pharmaceutical field; open museum expositions devoted to the problem of healthcare; conduct master class lectures and educational events devoted to the problem of forming a comprehensively developed personality; involve students in joint research on the problem of active longevity; provide informational support to educators and the public through scientific and popular science publications; conduct awareness-raising work with the population in order to create a health-saving environment.

## 1. Introduction

In the modern period, in order to achieve social well-being, there is an urgent need for different segments of the population to have social partnerships and obtain educational information from the scientific community, as well as medical and pharmaceutical professionals.

The need to create a healthcare system in Ukraine focused on people, results, and implementation is declared in the “National Strategy for Building a New Healthcare System in Ukraine for the period of 2015–2025” [1]. The role of the academic staff in medical and pharmaceutical education institutions in implementing the National Drug Policy in Ukraine, taking into account the context of the community, as well as conducting preventive work and providing professional advice and pharmaceutical care to the population, is growing.

It is important to study the influence of pharmaceutical scientists on the formation of a healthy lifestyle within the general population. This problem is particularly relevant in the context of the socio-economic crisis associated with the COVID-19 pandemic, quarantine restrictions, the mass movements of adults and children in Ukraine from temporarily occupied territories, and changes in the usual mode of activity of educational institutions.

The aim of this article is to analyze the role of Slobozhanshchyna’s pharmaceutical scientists in the formation of a healthy lifestyle as a value orientation of students and ordinary citizens in historical retrospect.

## 2. Materials and Methods

Historical research methods (chronological, logico-historical, retrospective) allowed us to substantiate the territorial and chronological boundaries of the study, providing us with a dynamic consideration of the subject of research. The territorial boundaries of the study cover Slobozhanshchyna, a historical and geographical region centered in Kharkiv (Kharkov in the Imperial era). The chronological boundaries of the study are determined by the emergence of the traditions of pharmaceutical scientists as they promoted the formation of a healthy lifestyle in the Sloboda region in the 19th and early 20th centuries, as well as the revival of the best of their traditions in the modern period. Attention to the educational activity of modern Slobozhanshchyna pharmaceutical scientists is determined by the fact that, at the present time, when a restructuring of pharmaceutical higher education in Ukraine is taking place, the traditions of pharmaceutical scientists as they relate to socially useful activities are being revived.

General scientific research methods (the analysis, synthesis, comparison, classification, systematization and generalization of legal documents, scientific publications and historical and reference publications) were used throughout the study. They allowed us to characterize the basic concepts of the research.

The retrospective method of research allowed us to focus on the approaches of the founders of pharmaceutical education in Ukraine to the problem of the formation of a healthy lifestyle as a value orientation, in historical retrospect, that has special scientific and practical value for the present.

This personalized method of research led to a careful study of the biographies of Slobozhanshchyna’s medical scientists and naturalists. This method allowed us to concentrate our attention on the life stages of scientists in which they were actively attempting to promote a healthy lifestyle as a value orientation of students and ordinary citizens.

The retrospective and personalized methods of research were used in the following research sources:–Speeches, reports, and historical essays of scientists (L. Vannoty, G. Lagermark) [2,3,4];–Historical and reference publications regarding the history of the Kharkov University Medical Faculty in the first hundred years of its existence (1805–1905) [5];–Documents from regional public and educational societies (the 19th to early 20th cent.), especially of the Kharkov Medical Society and the Kharkov Pharmaceutical Society, in the periodicals of the Imperial era («Хаpькoвcкий мeдицинcкий жуpнал» (“Harkovskiy meditsinskiy zhurnal”) [6], and newspapers «Южный кpай» (“Yuzhnyj kraj”) [7] and «Хаpькoвcкиe губepнcкиe вeдoмocти» (“Harkovskie gubernskie vedomosti”) [8]);–Data on the exhibition and museum activity of medical scientists and naturalists of the Imperial Kharkov University [9,10];–Visual sources (photos, certificates from the private collections of the pharmaceutical scientists of the National University of Pharmacy (NUPh));–Interpretive and research sources (theses, articles, monographs of leading scientists related to the research topic [11,12,13,14,15,16,17,18]).

The empirical research method (questioning and analytical processing of questionnaires) was used for the questioning of pharmacist interns on the problem of forming a healthy lifestyle involving researchers and the teaching staff of the NUPh, as well as analytical processing of questionnaires. In the period from June to October 2021, we conducted a study based on a questionnaire developed by the authors of the article among pharmacist interns of the NUPh (Kharkiv). The purpose of the survey was: (1) the analysis of the peculiarities of the formation of a healthy lifestyle of the future students specialists in the pharmaceutical industry; (2) giving recommendations for improving the effectiveness of health-saving measures in the University educational environment, (3) the substantiation of the role of the academic staff in involving pharmacy professionals to change the behavior of ordinary citizens in relation to their own health.

It should be noted that during the survey period, pharmacist interns already had a higher pharmaceutical education. The questionnaire was sent out using the Google form and was a list of questions and statements in the form of a questionnaire ordered by content and form. At the same time, it was possible to practically avoid the possibility of the subjects’ influence on each other and prevent the elements of suggestion.

The total number of respondents was 231 persons, including 216 females and 14 males. One person did not specify gender. Most of the applicants for postgraduate education who took part in the survey graduated from the NUPh. The sample was representative for pharmacist interns, future specialists in the pharmaceutical industry. The questionnaire work had appropriate oversight and vetting by officials: Dean of the Faculty of Advanced Training, Dean of the Faculty of Postgraduate Study, Director of the IATPS. The specificity of working with the questionnaire is shown in the explanation of its criteria:–Design: target group: pharmacist interns, applicants for postgraduate pharmaceutical education (about 2000 people);–Informed consent process: purpose of the survey was defined as analysis of the peculiarities of the formation of a healthy lifestyle in the institution of higher pharmaceutical education, outlining recommendations for improving the effectiveness of health measures on the basis of the institution of higher pharmaceutical education;–Development and pre-testing, survey administration: the questionnaire was developed by a team of authors and formed into a Google form, after which a QR code was generated. It was made public in the classroom for pharmacist interns who studied and entered the internship at the NUPh. They were asked to take the survey, their purpose and the estimated time of the survey—12 min (0.5 min for 1 question). With the help of the Internet and smartphones, pharmacist interns followed the QR-code to the questionnaire (Google-form) and passed the survey. It was a voluntary survey;–Time/date: questionnaire was created in June; the survey was conducted in 2 stages: the first one-in 22–23 of September 2021-among pharmacist interns who were already studying at the NUPh internship at that time (the respondents were 12 persons); the second one-since 27 of September till 4 of October 2021-among pharmacist interns who entered the internship of the NUPh (219 people became respondents);–Number of items: the questionnaire contained 24 elements;–Number of screens (pages): the questionnaire was placed on 4 screens;–Review step: respondents could view and change their answers (using the «back» button).

The questionnaire consisted of statements reflecting approaches both to the definition of the concept of a “healthy lifestyle” and to the ways to form a healthy lifestyle with the participation of pharmaceutical scientists and pharmacy professionals in the educational space of the University and beyond.

## 3. Results

### 3.1. The Terminological Field of Research

To analyze the problem outlined, it is important to define the terminological field of research. The concept of “value” is general scientific. According to the definition of the American Scientist M. Rokech, value is a stable confidence [11]. Values can be the subject of a person’s desire, interest, or need. According to a number of Ukrainian scientists in the field of educational, pedagogical sciences, this concept is defined as “a socio-philosophical category that records the positive or negative meaning of an object or phenomenon” [19] (p. 340).

Since the second half of the 20th century the concept of “value orientations” entered the scientific community. Value orientations reflect the degree of reflection of values in the activity. The system of stable value orientations of a person is an indicator of what can be expected from the individual. Therefore, we believe that value orientations should be interpreted as a selective attitude to the totality of material, social, and spiritual goods and ideals considered as objects of purpose, and it means to meet the needs of a person or a social group.

It is generally accepted that an important vital value of an individual is health. The World Health Organization (WHO) website states that health should be a fundamental human right, and all people around the world should have the highest attainable standard of health [20]. Numerous data of domestic and foreign scientists confirm that health depends on the lifestyle by 50–55% [12].

The world-famous surgeon, thinker M. Amosov (1913–2002) noted that the most complete relationship between the lifestyle and health is reflected in the concept of a “healthy lifestyle” [13] (pp. 63–70). At the same time, it should be mentioned that a “healthy lifestyle” is defined as a multifunctional directed activity that is interrelated with physical, somatic, mental and moral health, and requires a comprehensive approach; the human life activity, which is based on the principles of the reasonable organization of life, harmonious unity of physical, mental and spiritual functions of a person and thus provides an opportunity to preserve and strengthen health; a full and productive life; achievement of active longevity [21] (p. 165).

The formation of a healthy lifestyle as a value orientation of an individual is a long process, and it is mediated by a number of factors, including: (a) circumstances of the individual’s life (conditions of education of the individual); (b) individual consciousness reflecting the general and specific conditions of the personality formation; (c) motivation of the real behavior mediated by the subject’s activities.

### 3.2. Training of Pharmacists in the NUPh in the Context of the System of Pharmaceutical Education in Ukraine

It is useful to analyze the factors affecting the formation of a healthy lifestyle as a value orientation in the educational space of the Slobozhanshchina higher education institution where pharmacy specialists are trained. It is important to briefly describe the system of higher pharmaceutical education in modern Ukraine.

Today in Ukraine professionals in specialties «Pharmacy», «Technology of Pharmaceuticals», «Clinical Pharmacy», «Technology of Perfumes and Cosmetics» are trained mainly at the National University of Pharmacy, Pharmaceutical faculties of Medical Universities and Academies in many cities of Ukraine (Kharkiv, Vinnytsya, Dnipro, Zaporizhzhia, Ivano-Frankivsk, Lviv, Kyiv, Odesa, Ternopil, Chernivtsi). According to the Ukrainian system, graduates cannot commence work in pharmacies without the Certificate of internship.

The main task of the internship is to increase the level of practical and theoretical training of pharmacist interns, their professional readiness for the independent activity in the specialty. The internship program consists of two parts: educational and practical. The practical part of the internship is conducted at the premises of pharmacies within 7–8 months. Then, the educational part of the internship begins in an institution of higher or postgraduate education, which trains pharmacist interns. The duration of the educational part of the internship is three months [22].

The center of the development of pharmaceutical education and science in Ukraine is the National University of Pharmacy. Geographically, the NUPh is located in the Eastern Ukraine, in Slobozhanshchina, in Kharkiv.

NUPh scientists have created 161 drugs. There are three specialized scientific councils working in the following fields of scientific specialties: “Pharmaceutical Chemistry and Pharmacognosy“, “Pharmacology“; “Technology of Drugs, Organization of Pharmaceutical Business and Forensic Pharmacy“, “Standardization and Organization of Drug Manufacturing“. In the structure of the University, except for 4 faculties (40 departments), there are also other structural divisions [23].

The Institute for Advanced Training of Pharmacy Specialists (IATPS) at the NUPh gives the opportunity to organize the educational part of the internship. Here, the pharmacist interns have classes (lectures and practices) and the «Krok-3» examination. The «Krok 3» examination assesses the compliance of the professional competence level of qualified specialists. According to the current regulations, this examination is a mandatory component of the Certificate of internship. It is organized by the Ministry of Health of Ukraine. After «Krok 3», pharmacist interns pass the examination in the IATPS and get the Certificate of internship as a license for work in the Ukrainian pharmacies.

In the focus of attention of the NUPh pharmaceutical scientists and students are efforts to respond to challenges of the time, be an example of hygiene, general social culture, demonstrate the need for a healthy lifestyle. This institution of higher pharmaceutical education became the initiator and the base of National Congresses of pharmaceutical workers of independent Ukraine, initiated the introduction of a professional holiday—the Day of a Pharmacist of Ukraine (every third Saturday in September) and the title of Honored Worker of Pharmacy of Ukraine [14]. The “NUPh on the side of health” slogan has become widespread in the educational space of the NUPh and has become the University’s task.

An important way to implement this task is the willingness of teachers of the NUPh, including pharmaceutical scientists, to introduce health-saving training technologies and create a health-saving educational environment.

Circumstances of life of future pharmacy specialists, their individual consciousness, and activities to maintain a healthy lifestyle largely depend on the corporate culture of the NUPh. One of the strategic goals of the NUPh is to create a student-centered space at the University [24]. This goal provides for increasing the social responsibility of the University for the formation of value orientations of future professionals, in particular maintaining a healthy lifestyle.

It should be noted that the mission of the NUPh is to provide knowledge and develop skills among students and pharmacist interns at the level of modern world standards; the vision is to train the skilled personnel for pharmacy. The philosophy of the NUPh is embodied in the slogan “Education with traditions” [24]. That is why the corporate culture of the NUPh is based on the principles of humanism, democracy, solidarity, unity of science and practical action professed and defended by founders of medical and pharmaceutical education in Ukraine (the 19th–early 20th centuries). It was found that among the founders of medical and pharmaceutical education in Ukraine there were medical scientists and naturalists of the Imperial Kharkov University (1804–1917) [16], Kharkiv Chemical and Pharmaceutical Institute (1921–1924) [14,15].

It is important for improving the competence of NUPh pharmaceutical scientists to get additional psychological and pedagogical education. Thus, a system of methodological assistance to improve the pedagogical skills of teachers has been proposed in the NUPh. This need is fulfilled by the generalization of the theory and practice of the teachers’ work on the use of modern educational technologies (in particular, health-saving training technologies) in the IATPS advanced training courses. The content of training, by the way, involves mastering the methods of implementing health-promoting educational technologies. The content of the educational process is organized in such a way that teachers based on theoretical developments and their own practical experience are able to correct the worldview of young people, form a health-saving educational environment.

Thus, the IATPS proposes advanced training courses for teachers from different pharmaceutical faculties. At these courses the pharmaceutical scientists in the context of analyzing the components of the teacher’s pedagogical culture discuss with interest the features of pedagogical, public, and educational activity of scientists of the past centuries. The analysis of the role of Slobozhanshchyna pharmaceutical scientists in the formation of a healthy lifestyle as a value orientation during the Imperial era is of particular interest.

### 3.3. The Public and Educational Activity of Slobozhanshchina Scientists (the 19th–Early 20th Centuries)

The analysis of research sources gave us the opportunity to summarize some important examples of socially useful activities of the founders of pharmaceutical education in Ukraine (especially medical scientists and naturalists of the Imperial Kharkov University). These examples given in Table 1 indicate the desire of scientists to shape a healthy lifestyle as a value orientation of students and ordinary citizens.

The organization of the educational work among the population was facilitated by the participation of the founders of pharmaceutical education in the work of the Kharkov Medical Society and the Kharkov Pharmaceutical Society.

The Kharkov Medical Society was a connecting link between Slobozhanshchyna doctors, pharmacists, and teachers of the Medical Faculty of the IKhU. Having started its work in 1861, the Society attracted public attention to pressing health issues. A functional diagram of forms and methods of the Society is presented in Figure 1.

An Honorary member of the Kharkov Medical Society was Ilya Ilyich Mechnikov (1845–1916), who received the Nobel Prize in Physiology or Medicine (1908). It was proven that the members of the Kharkov Medical Society were interested in both I. Mechnikov’s scientific activity at the Pasteur Institute in Paris and in his works, which had philosophical and ethical character (“Sketches of Optimism”, etc.) [17]. Let us pay attention to the fact that for I. Mechnikov the connection between the formation of the ability to an optimistic perception of life and active longevity was obvious. I. Mechnikov recommended teachers remember that the sense of life could be developed; to direct the education process towards an optimistic outlook. Nowadays, it is proven that the psychosocial core formed in childhood and youth is the inner essence of a personality, forming the basis of values.

The Kharkov Pharmaceutical Society started its work in 1881. It is significant that the Society focused its activity on organizing leisure for pharmacists and pharmacy workers. For this purpose, the project of a Pharmaceutical club was discussed in 1910 [8]. According to the initiators of the institution, the club should have an educational character. The following forms of the educational activities of the institution were provided: the arrangement of a library and a reading room, conducting readings and lectures on issues that were important for a pharmaceutical professional [16] (p. 621). The achievement of the Kharkov Pharmaceutical Society in the field of education was the foundation of a “new cultural institution”, the Institute, in 1913. The board of the Institute included: Director, organic chemist, Doctor of chemical sciences, Professor K. Krasusky (1867–1937), members of the board, Master of Pharmacy J. Zilberg and Doctor M. Ryasnyansky. The program of activities of the new Institute was mainly as follows:

checking and determining the quality of all drugs supplied to pharmacies of both foreign and domestic production;the study of the action of pharmaceuticals;the fight against falsification in all its forms in the supply of medicines;provision of the population, the opportunity to protect themselves from the use of spoiled and harmful food and products of economic and technical production, such as milk, butter, wine, water, beer, canned food, lemonade, kvass, flour, bread, wax, wallpaper, children’s toys, etc.techno-chemical analyses: studies of ores, alloys, coal, oil, minerals, soil, agricultural fertilizers, and fodder for cattle and various studies of bacteriological and chemical and microscopic character.

The founders of pharmaceutical education in Ukraine took part in exhibition activities, organization of museum expositions to promote scientific achievements among the public. Andriy Krasnov (1862–1915), the well-known geographer and botanist, professor of the IKhU took part in the creation of the ethnographic exhibition at the 12th Archaeological Congress (Kharkov, 1902) [10]. On the basis of this exhibition, the museum was founded. The demonstration of the traditional medicine collection in the museum helped to spread information about the ways to prevent diseases. The collection was housed in four large windows at the IKhU and consisted of medicinal herbs (more than two hundred) with the name of each and indications of what diseases they helped [9] (p. 221).

It should be noted that during the 19th–early 20th centuries the founders of pharmaceutical education, professors of the IKhU, Kharkiv Chemical and Pharmaceutical Institute left a noticeable mark in the retransmission of the latest scientific knowledge on a healthy lifestyle from the Western European cultural centers to Slobozhanshchyna.

The analysis of historical sources suggests that pharmaceutical scientists were able to make a contribution to the achievement of social well-being. Medical scientists and naturalists of the Imperial Kharkov University put forward ideas and carried out activity in relation to healthy lifestyle formation among students during both the academic work and the extracurricular time (during the University solemn acts, meetings of the Medical Faculty Council, the Society of Research Sciences at the IKhU; through demonstrations of exposures on health protection in museums and at exhibitions, etc.). The educational activities of Slobozhanshchyna pharmaceutical scientists in forming a healthy lifestyle were driven by the organization of the Kharkov Medical Society, Kharkov Pharmaceutical Society.

Nevertheless, Kharkov scientists were unable to implement all their ideas regarding the formation of a healthy lifestyle in the 19th and early 20th centuries. Most of the issues related to the formation of a health-saving environment and a healthy lifestyle of the population could not be solved due to the lack of favorable socio-political conditions.

It is important for modern pharmaceutical scientists to ensure an effective partnership with students, pharmacist interns, and the public in order to develop the value orientations of a healthy lifestyle at the regional level. Such a partnership should be based on the common interests, mutual respect, account the best traditions of the founders of pharmaceutical education. Our experience shows the expediency of the introduction of such ways of educational work with the public: edutainments (scientific picnics), publications on the effectiveness and safety of dietary supplements, the study of foreign experience in the organization of lifelong learning, etc.

An important condition to conduct the educational work among students was future pharmacists is creating a health-saving educational environment at the University.

### 3.4. The Role of Slobozhanshchyna Pharmaceutical Scientists in the Formation of a Healthy Lifestyle as a Value Orientation of Students and Ordinary Citizens in the Current Conditions

Nowadays, at the premises of the NUPh, the study and creative use of pedagogically valuable theoretical developments and practical achievements of the founders of pharmaceutical education in Ukraine in forming a healthy lifestyle by means of education and science continue.

Indicators of creating a health-saving educational environment are as follows: improving the physical performance and mental capacity of students and teachers; improving the competence of teachers in healthcare matters and developing their willingness to form the value orientation of future pharmacy specialists to a healthy lifestyle; tracking the dynamics of morbidity of students and employees of the University, which indicates the final result of the formation of a healthy lifestyle; reducing the level of anxiety and preventing the emotional burnout of students and the University staff (the Psychological service functioning); participation of students in the health-saving activity.

At the premises of the NUPh, there is a museum with exhibitions on the history of higher pharmaceutical education. The materials of the expositions motivate teachers and students to search for creative approaches to organizing the professional activity in the field of healthcare.

In order to provide the information support to teachers and students on self-improvement on the problem of a healthy lifestyle, the University publishes scientific and popular science literature, creates an electronic database of publications, develops electronic courses, introduces psychological and pedagogical subjects for future pharmacy specialists, activates the volunteer spirit, organizes the Center for Pharmaceutical Local History, etc.

Our experience shows the effectiveness of the promotion of a healthy lifestyle by teachers through popular lectures, explanatory work with the residents of student dormitories on hygiene and sanitation. Master class lectures have become a tradition at the NUPh. Their content is interdisciplinary in nature, covers the topical issues of pharmacogenetics, gender pharmacology, and concerns the problem of overcoming alcohol, tobacco, and drug addiction.

Among many famous traditions developed over the centuries a special place is occupied by a deep understanding of the importance of the educational activity by pharmacy teachers among the general population in the context of implementing the idea of a healthy lifestyle. An interesting form of the educational work is scientific picnics for residents of Kharkiv.

“Scientific Picnics”: Teachers and students in tents, squares, or other public areas of the city, together with ordinary citizens, combine science with fun, explaining and demonstrating the possibilities of chemistry, biology, pharmacognosy, and other fields of science in creating a health-saving environment (Figure 2).

The main idea of research experiments in the framework of “Scientific Picnics” is to explain, at first glance, complex things in simple words. Various Universities and Research Institutes of the Kharkiv region are involved in this event. The NUPh has repeatedly participated and continues to take part in the scientific picnics—the largest open-air Science Festival in Ukraine, traditionally held in the second half of September, covering its scientific and educational activities. Depending on their scientific field pharmacists acquaint everyone with the variety of medicinal plants, the magical world of microorganisms, refute the popular myths about the mechanism of action of well-known medicines and demonstrate methods available to everyone to check the quality of drugs and food products. Scientists are implementing a bright interactive program for visitors of the event lasting for about 4 h.

The educational activity of modern pharmaceutical scientists on dietary supplements: To form a healthy lifestyle, it is important to train people at any stage of their life path. The educational activity of modern pharmacists helps to prevent diseases, strengthen all body systems, and improve the human general well-being. The lifestyle has the greatest impact on health, and a healthy lifestyle, in turn, lays the foundation for good health, strengthens defenses of the body, and reveals its potential.

It is known that nowadays nutrition often does not meet the needs of the body of an active person, athletes, pregnant women, children, or the elderly by the content of biologically active substances. Modern science gives answers to many questions about the amount of biologically active substances, their composition for enriching the diet. Dietary supplements (DS) are becoming more and more popular all over the world. However, it should be noted that DS should be of good quality, do not contain harmful or potent substances, and their use should serve to preserve human health. There is also a threat that DS may interact with medications and reduce or potentiate their effects [25,26,27,28,29,30].

When solving the issue of quality, effectiveness, and safety of DS, it is necessary to combine the efforts of manufacturers, distributors, public health agencies, enforcement officials, and other organizations, invest in organizing educational activity for the safe and useful use of these products [25,26,27,28,29,30]. Undoubtedly, a great role in solving this question belongs to pharmaceutical scientists. The world’s pharmacopoeias include monographs on DS [31,32]. Students studying at pharmaceutical faculties are offered various courses that introduce them to various issues related to a healthy lifestyle and the use of DS for maintaining health.

International Pharmaceutical Summer Camp: In particular, the international event “The 15th International Pharmaceutical Summer Camp”, whose topic was “Food supplements”, was organized at the Faculty of Pharmacy of the University of Ljubljana (Slovenia, 2011). The NUPh students took part in the event. The purpose of the event was to provide future pharmacists from different countries (France, Serbia, Slovenia, Romania, Portugal, and Ukraine) with knowledge about the legal framework for the use, history of DS creation, medicinal plants in their composition, the results of scientific work, quality control, distribution, and marketing of DS. Students listened to lectures by professors from the Faculty of Pharmacy of the University of Ljubljana and the NUPh, met with the leading manufacturer of DS, who introduced their range of DS, the head of the pharmacy where DS was sold. Moreover, the students themselves harvested the medicinal plant raw material and made DS under the supervision of professors of the Faculty of Pharmacy of the University of Ljubljana (Figure 3 and Figure 4). Thus, students gained experience in production, making recommendations for the use of DS. The acquired knowledge, skills, and abilities proved to be useful for maintaining the health of future pharmacists, as well as for forming a value orientation regarding the organization of a healthy lifestyle of ordinary citizens.

Nowadays, pharmacist interns as applicants for postgraduate education at the premises of the IATPS NUPh must not only be trained for the active professional activity, but also be able to immediately adapt to a certain situation, improve personal qualities, demonstrate general education and culture, sociability, independence in decision-making and responsibility for their actions, and contribute to the achievement of social well-being.

### 3.5. The Analysis of the Peculiarities of the Healthy Lifestyle Formation in the Institution of Higher Pharmaceutical Education

At the final stage of the study, we processed, analyzed, and interpreted the data obtained. The results of the analysis of the questionnaires showed that majority of the respondents revealed the professional and individual formation of the components of the health culture: literacy and willingness to follow a healthy lifestyle. In the opinion of 199 (86.1%) respondents, the concept of a “healthy lifestyle” included the cessation of smoking and alcohol consumption, sports, physical exercises, prevention of emotional burnout, the rational work and recreation system, and healthy eating. It is significant that 119 pharmacist interns (51.5%) showed a positive attitude to vaccination, considering it in the context of a healthy lifestyle.

The analysis of the responses allowed us to state that the overwhelming majority of respondents (61.0%) classified themselves as those who led a healthy lifestyle. Respondents (67.1%) showed that the example of well-known pharmaceutical scientists was an important factor in the formation of a healthy lifestyle of students. Respondents (58%) pointed out the importance of the role of the University teachers in the formation of a healthy lifestyle for students (Figure 5).

At the same time, all respondents as specialists in the field of healthcare considered it necessary to conduct active public and educational activities among the population (Table 2).

Therefore, we can state that the priority given by respondents to a value orientation towards a healthy lifestyle is accompanied by their recognition of the need for its active transmission among ordinary citizens during their professional activities. Moreover, pharmacist interns recognized the significant role of the particular pharmaceutical scientists, in the formation of a healthy lifestyle of future pharmacy professionals in the University space.

The analysis of the survey results revealed that the respondents’ awareness of the value orientation to a healthy lifestyle was due both to the content of thematic lectures, master classes, excursions to museums (Anatomical Museum of the Kharkiv National Medical University, NUPh museum, V. N. Karazin Kharkiv National University Nature Museum, Pharmacies-museums, etc.), drug dispensaries, arboretums, and botanical gardens, and the creation of a health-saving environment in the educational space (introduction of health-saving educational technologies; real, consistent implementation of the program of forming a healthy lifestyle of young people).

Our data show that, as a result of comparing the relevance of approaches to the formation of a healthy lifestyle for future pharmacy students, respondents assessed their dominance and ranked them from the most to the least relevant. The majority of respondents did not find it possible to recognize any of the approaches proposed as irrelevant (Table 3).

Based on the analysis of the results of the questionnaire, we have formulated the indicators showing the respondents’ awareness of the value orientation to a healthy lifestyle. These indicators include:
knowledge (about health as a global phenomenon, the consequences of bad habits, etc.);success (having life skills in planning and decision-making regarding a healthy lifestyle, means of maintaining the emotional health, effective communication);positivity (positive emotional reactions in the perception of knowledge about the preservation and promotion of health; satisfaction in the performance of health development activities);manageability (manifestation of strong-willed efforts to participate in events, the implementation of activities for the formation of a healthy lifestyle; willingness to defend life principles while exerting some pressure; readiness for self-development;meaningfulness (the ability to predict and assess the consequences of activities related to one’s own health and the health of other people, assessment of the emotional state, the emotional experience and health activities; assessment of the effectiveness of activities aimed at preserving and developing health).

All respondents noted that special events were held in the University to form a healthy lifestyle and pointed out those that they personally witnessed or participated in (Table 4).

The questionnaire revealed the respondents’ assessment of the effectiveness of the measures implemented to form a healthy lifestyle at the University. Using a scale from 1 to 10 (1—the minimum efficiency indicator, 10—the maximum efficiency indicator) the majority of respondents chose the number 8 on the scale. This result of the survey indicates a high assessment of the effectiveness of the activities in the health-saving educational environment.

A survey of pharmacist interns at the premises of the IATPS NUPh using a questionnaire has revealed the presence of indicators showing the respondents’ awareness of the value orientation to a healthy lifestyle. Among these indicators, there are awareness; progress; positivity; manageability; sense of purpose. It has been found that the example of well-known pharmaceutical scientists and teachers of the future student specializing in pharmacy is an important factor in the formation of a healthy lifestyle of pharmaceutical education applicants.

## 4. Discussion

The role of Slobozhanshchyna pharmaceutical scientists in the formation of a healthy lifestyle as a value orientation of students and ordinary citizens during the 19th and early 20th centuries, as well as in the modern period, is presented. It was mentioned that the “healthy lifestyle” was defined as a multifunctional directed activity that had interrelated with physical, somatic, mental, and moral health. It was noted that the founders of pharmaceutical education in Ukraine initiated the educational activities in forming a healthy lifestyle in the educational space and outside of the Imperial Kharkov University. The main results of independent studies of scientists on a healthy lifestyle were reflected both in publications of scientific and popularized nature.

It has been proposed to define the concept of “orientation to a healthy lifestyle of representatives of the institution of higher pharmaceutical education”. It includes the selective attitude of employees and students to a healthy lifestyle as a set of material, social, and spiritual benefits and ideals considered as objects of purpose (implementation of the mission of the University) and tools to meet the needs of both University representatives and ordinary citizens. The formation of a healthy lifestyle as a value orientation of an individual is a long process, and it is mediated by a number of factors. Among them: (a) circumstances of the individual’s life (conditions of education of the individual); (b) individual consciousness reflecting the general and specific conditions of the personality formation; (c) motivation of the real behavior mediated by the subject’s activities. The presence of these factors is typical for the health-saving educational environment.

The role of pharmaceutical scientists in forming an environment for the development of a harmonious, spiritually and physically developed personality is important. The pharmaceutical scientists as the teachers of pharmaceutical disciplines will have the opportunity to get acquainted with materials in the areas of professional, research, public, and educational activities of the founders of pharmaceutical education. Such information would be introduced to the content of the subjects of advanced training courses.

Thus, our study shows that among the important indicators of creating a health-saving educational environment there are the following ones: improving the physical performance and mental capacity of students and teachers; improving the competence of teachers in healthcare matters and developing their willingness to form a healthy lifestyle in the University space; the introduction of health-saving educational technologies in the University space; tracking the dynamics of morbidity of students and employees of the University, which indicates the final result of the formation of a healthy lifestyle; reducing the level of anxiety and preventing the emotional burnout of students and the University staff (the Psychological service functioning); participation of students in health-saving activities.

## 5. Conclusions

In modern times, pharmaceutical scientists of the NUPh are trying to organize a socially useful activity taking into account the best achievements of the founders of pharmaceutical education in Ukraine. The educational activity of the NUPh scientists is aimed at training future pharmacy specialists as professionals and socially responsible citizens, conducting research in accordance with the main global trends in the development of healthcare, and providing expert and advisory services to civil society and the authorities. An important direction in implementing the mission, vision, and strategic goal of the NUPh is to involve scientists in achieving social well-being and socio-economic development of the region and educating the society through the formation of a healthy lifestyle as a social orientation.

To organize effective promotion of a healthy lifestyle by modern pharmacists scientists, we can recommend the following measures: providing an opportunity to study the peculiarities of organizing the educational activity of scientists of the past to students and specialists in the pharmaceutical field; opening of museum expositions devoted to the problem of healthcare at the premises of pharmaceutical education institutions; conducting master class lectures and educational events devoted to the problem of forming a comprehensively developed personality; involvement of students to joint research on the problem of active longevity; providing information support to educators and the public through scientific and popular science publications; conducting the awareness-raising work with the population in order to create a health-saving environment.

The presence of indicators showing the awareness of pharmacist interns about the value orientation to a healthy lifestyle formed, among other things, under the influence of the pedagogical, educational activities of pharmaceutical scientists has been found. These indicators include: knowledge about health as a global phenomenon; success when forming life skills in planning and decision-making regarding a healthy lifestyle, means of maintaining the emotional health, effective communication; positive emotional reactions in the perception of knowledge about the preservation and promotion of health; satisfaction in the performance of health development activities; manageability as manifestation of strong-willed efforts to participate in events, the implementation of activities for the healthy lifestyle formation, readiness for self-development; meaningfulness as the ability to predict and assess the consequences of activities related to one’s own health and the health of other people, assessment of the emotional state, the emotional experience, and health activities.

Thus, this article allows us to conclude that the role of Slobozhanshchyna pharmaceutical scientists in forming a healthy lifestyle as a value orientation of students and ordinary citizens in historical retrospect is really important. Nowadays, an active participation of pharmaceutical scientists in the creation of a health-saving educational environment will allow for the formation of a healthy lifestyle as a value orientation of students and pharmacist interns. It would be important for enhancing the pharmacists’ educational activity among ordinary citizens.

The research conducted does not fully cover all aspects of the scientific problem raised. The analysis of the educational activities of Soviet pharmaceutical scientists is necessary. Further research is required in the field of the introduction of health-saving educational technologies in the Institute for Advanced Training of Pharmacy Specialists (IATPS) at the National University of Pharmacy (Kharkiv) in the context of the COVID-19 pandemic quarantine restrictions.

## Figures and Tables

**Figure 1 pharmacy-10-00020-f001:**
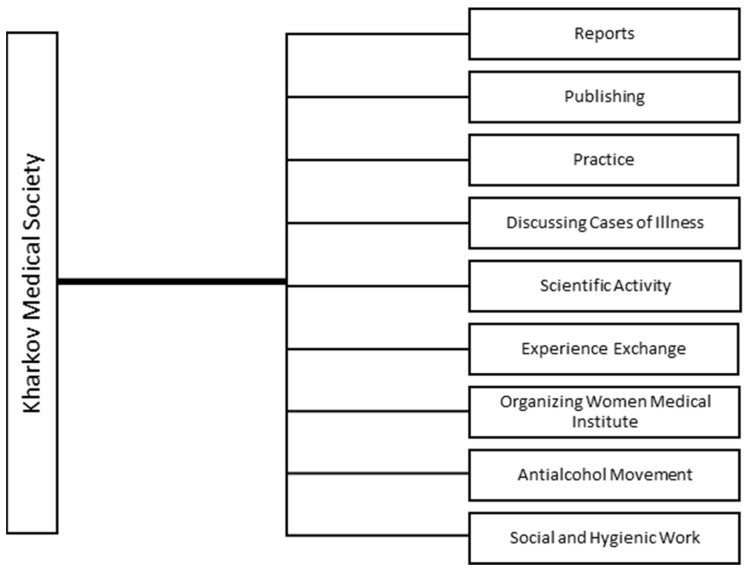
A functional diagram of forms and methods of the Kharkov Medical Society [16] (pp. 456–460), [17].

**Figure 2 pharmacy-10-00020-f002:**
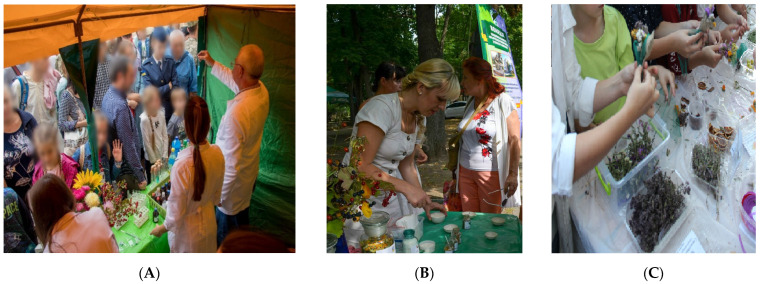
(**A**) Assoc. prof. of the NUPh Volodymyr Hrudko demonstrated the possibilities of Chemistry for ordinary citizens; (**B**,**C**). Teachers of the NUPh Helena Rudakova and Olga Rudakova conducted a master class on making boutonnieres from medicinal plants.

**Figure 3 pharmacy-10-00020-f003:**
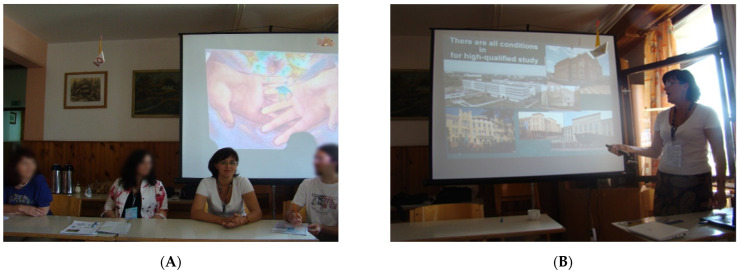
(**A**) A round table discussion on the topic of food supplements; (**B**) Lenchyk L., professor of the NUPh gives a lecture at the Summer Camp.

**Figure 4 pharmacy-10-00020-f004:**
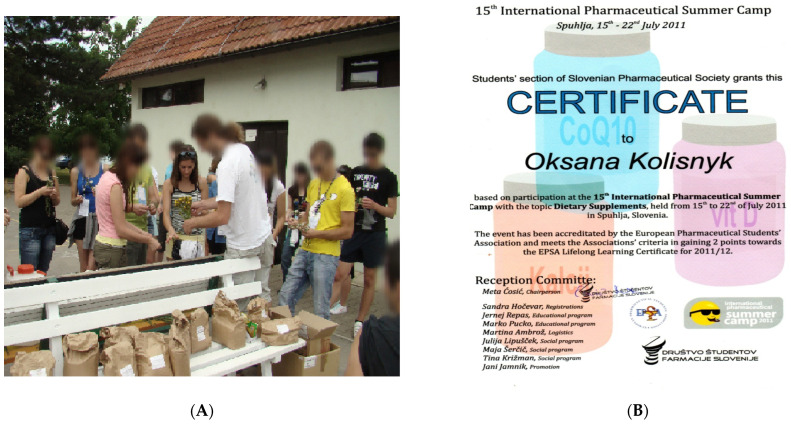
(**A**) Student of the NUPh Oksana Kolisnyk makes DS under the supervision of the professor in an international group of future pharmacists; (**B**) A certificate received by the participants of the event.

**Figure 5 pharmacy-10-00020-f005:**
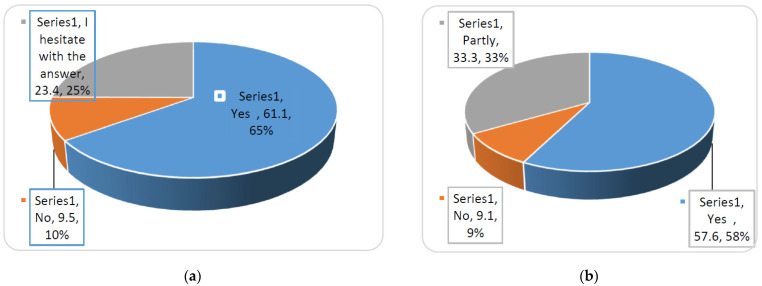
Respondents’ answers about the importance of the example of teachers in a healthy lifestyle. (**a**) In your opinion, is the example of well-known pharmacists an important factor in shaping a healthy lifestyle of the student youth? (**b**) Are the teachers of the University an example for you in the formation of a healthy lifestyle?

**Table 1 pharmacy-10-00020-t001:** The examples of socially useful activities of the founders of pharmaceutical education in Ukraine.

Medical Scientists and Naturalists of the Imperial Kharkov University	The Examples of Socially Useful Activities
Ferdinand Giese (1781–1821)—professor of Chemistry, Doctor of Philosophy, a corresponding member of the Imperial Academy of Sciences in St. Petersburg and the Medical and Surgical Academy of Russia	The part in the discussion about studies of the regional water resources during the meetings of the Boards of the Medical Department of the IKhU, a report on the results of the chemical analysis of the pond water of the Kochubey estate (1807) [5].
Grigory Koritari (1772–1810)—professor of the Department of Medical Rhetoric, Pharmacy and Medical Literature (Doctor of Medicine and Master of Ophthalmology)	The participation in the discussion about studies s of the pond water of the Kochubey estate (Slobozhanshchyna) during the meetings of the Board of the Medical Department of the IKhU, comments on the therapeutic effect and usefulness of this water (1807) [5] (p. 329).
Ludwig Vannoty (1771–1819)—professor of the Department of Medical Literature, Pharmacy and Medical Literature	The Speeches “About Horrible Pollution of Kharkov, which Causes Various Diseases“ at the meeting of the Medical Faculty Council of the IKhU (1807) [5]; “About amphibians, fish and the simplest insects that are found in the Kharkov province, the usefulness of such a list and its need for a physiologist” (1813) [2], “About the probable cure of almost all diseases” (1818) at the IKhU Solemn act [3] (p. 591).
Gregory Rindovsky (1814–1898)—professor, the first head of the Department of Theoretical and Experimental Pharmacology	The attempts to explain the therapeutic effects on the sick human body, to outline rational instructions for their medical use [16] (p. 317).
Dementiy Rodzaevsky (1857–1894)—Head of the Department of Pharmacology; Sergei Popov (1850—after 1919)—Doctor, pharmacologist, Honored Professor of the IKhU	The study of the healing qualities of mineral springs in the Caucasus [18] (pp. 29, 32).
Yegor Gordienko (1812–1897), a pharmacologist and pharmacist, Doctor of Medicine (1838), professor, public figure	The conducting of public readings, in particular on Chemistry, at the IKhU for representatives of all social classes (from 1838). An active participation in the organization of the Kharkov Society of Spreading Literacy among the people (from 1869) [16] (p. 601).
Andriy Chirikov (1849–1912)—professor, Head of the Department of Pharmacy and Pharmacognosy (IKhU), Master of Pharmacy (1883)	The report about his method of clarifying and purifying liquids with magnesium hydrate at the meeting of the Physico-Chemical section of the Society of Research Sciences at the IKhU (1880). Defense of the master’s thesis «Study of Coal» (1882) [16] (pp. 334, 337). The participation in the development of physicochemical methods of water purification (the first examples of using colloidal coagulants). Attention to the study of the history of the Kharkov water supply system. The publications of the research results of water sources [16] (p. 609). Public activity as a member of the Kharkov City Duma (1881–1890), the Board of Trustees of the city children’s outpatient clinic, the chairman of the commissions on the purchase of medicines for city hospitals and outpatient clinics [7]. Membership in the Kharkov Medical Society, Kharkov Pharmaceutical Society. The participation in the establishment of an analytical commission at the company, which was responsible for studying goods from local markets (from 1898) [16] (p. 448). The participation in the organization of the educational process at the Women’s Medical Institute of the Kharkov Medical Society (from 1910) [16] (p. 605). Membership in the first Board of the Institute [6].
Mykola Valyashko (1871–1955)—professor of the Department of Pharmacy and Pharmacognosy	Involvement in the work of the Kharkov Department of the Military Chemical Committee during the First World War, facilitation to the organization of research institutes in Ukraine, Kharkiv Chemical and Pharmaceutical Institute (1921) [16] (p. 291).

**Table 2 pharmacy-10-00020-t002:** Indicators for assessing the importance of the public and educational activity methods of pharmacy specialists in forming a healthy lifestyle among the citizens.

Methods of Information and Educational Activities of Pharmacy Specialists	Number (%) of Respondents who Considered this Method to Be Significant
Participation in the design of the windows of a particular pharmacy	4.3
Preparation and publication of health-related materials in journals, collections of scientific papers (conference proceedings, etc.), newspapers	29.4
Informing pharmacy visitors about the products available for disease prevention (vitamins, etc.)	22.1
Participation in organizing and functioning of public organizations of pharmaceutical workers	15.6
Publication of health-saving materials on social networks	12.1
Popularization of knowledge about the peculiarities of drug storage and cultivation of medicinal plants	10.4
Involvement in conducting lectures on health education	6.1

**Table 3 pharmacy-10-00020-t003:** Distribution of respondents’ approaches to the formation of a healthy lifestyle of the students-future pharmacy specialists by the level of relevance (dominance).

No.	Approaches to the Healthy Lifestyle Formation	The Most Relevant (%)	Rather Relevant (%)	The Least Relevant (%)	Not Relevant (%)
1	Involvement of scientists in the development of the state program for the formation of a healthy lifestyle of young people and its real, consistent implementation in the educational space during conferences, congresses, organization of the research work of students	58.0	39.0	2.2	0.9
2	Activities of the Center of Health, Sports and Recreation; the Cultural Center	56.3	41.1	2.6	0
3	Activities of the Psychological service of the University	45.5	42.9	9.5	2.2
4	Medical examinations of students and the staff of the University to assess the health status	63.6	31.6	4.3	0.4
5	Introduction of disciplinary punishments, up to expulsion, in case of detection of alcohol abuse and smoking by students	24.2	32.0	21.2	22.5
6	Visual propaganda about the consequences of bad habits and the ways to form a healthy lifestyle	38.1	46.3	12.1	3.5
7	Motivation of employees of the University and applicants for pharmaceutical education to follow the recommendations concerning a healthy lifestyle, in particular in the context of the COVID-19 pandemic and quarantine restrictions	45.9	42.9	8.2	3.0
8	Creation of student youth organizations for a healthy lifestyle	38.1	48.1	7.8	6.1
9	Creating an affordable infrastructure for a healthy lifestyle, in particular at the premises of the University	42.4	46.3	10.0	1.3
10	Introduction of benefits for the use of sports facilities/clubs at the premises of the University for students/youth	56.7	35.1	4.8	3.5
11	Involvement of student self-government bodies in the organization of health awareness building campaign	35.9	48.5	11.7	3.9

**Table 4 pharmacy-10-00020-t004:** Involvement of respondents in activities at the University aimed at forming a healthy lifestyle.

Special Measures for the Healthy Lifestyle Formation	Witnesses or Participants of the Event among Respondents (%)
Lectures on a healthy lifestyle, the consequences of bad habits	38.6
Prohibition of smoking	10.2
Availability of billboard campaign on the problems of the consequences of bad habits	2.3
Activities of the Center for Health, Sports and Recreation; the Cultural Center; the Psychological service	20.0
Conducting educational activities for the purpose of forming a healthy lifestyle	8.4
Discussion of the ways to form a healthy lifestyle during conferences, congresses and other educational and scientific events	19.5
All named options	1.0

## Data Availability

https://docs.google.com/forms/d/e/1FAIpQLSdqAj_ikiyQKbT3P16eckoxlGnbeR6uV_gh2v6b0d0ZAHaqWQ/viewform?usp=sf_link; https://docs.google.com/spreadsheets/d/19dCLv-zE1aELvc__NRppjeXcLJo2GI4Sy8IaKjZsQv0/edit#gid=1728763996.

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
