# Peer review of "The Role of Pharmaceutical Scientists in the Formation of a Healthy Lifestyle as a Value Orientation"

_pharmacy, 2022, doi:10.3390/pharmacy10010020_

Round 1

Reviewer 1 Report

Thank you for this revision.  I especially appreciated the construction of Table 2 and Figure 1 as a means to summarize some of the historical content.  The addition of Table 1 helps to quickly review your questionnaire.  The discussion of indicators is important.  I appreciated this being in the abstract and conclusion for emphasis.

A few minor edits:

-please remove "This allowed us to draw empirically sound conclusions" on line 488.  It could be argued that the response rate, etc. influenced the soundness of the conclusions.  The paragraph stands without this sentence and taking it out avoids you needing to declare limitations.

-This statement is confusing given Table 4: None of the respondents considered it possible to recognize any of the approaches proposed as
irrelevant (Table 4).  Consider deleting or modifying.  It seems that some DID mark "not relevant", so "none" doesn't make sense.

-Line 592.   Please replace "proven" with something such as "Our study indicates that...."  Proven is a very strong word.  We generally don't "prove" anything with one study.

The paper provides a nice balance of historical and current perspectives.  I appreciated the descriptions of events and the focus on the effect of scientists on a value orientation!  Thank you for your careful attention to the recommendations.

Reviewer 2 Report

This article is a narrative papers on the education programme conducted in Ukraine.

I have some major issues with this report

Abstract

 Abstract. The article focuses on the role of pharmaceutical scientists in achieving the social well-20 being, in particular in the formation of a healthy lifestyle as a value orientation among students and 21 the public in historical retrospect. The following set of research methods was used: general scientific; 22 historical (chronological, logico-historical, retrospective), personalized, empirical (questioning of pharma-23 cists-interns and analytical processing of questionnaires). The chronological methods allowed to con-24 sider the subject of research in dynamics and time sequence. The retrospective method allowed to 25 focus on those approaches of the founders of Pharmaceutical education in Ukraine to the problem 26 of formation of a healthy lifestyle as a value orientation that have special scientific and practical 27 value for the present. The personalized method of research allowed to concentrate attention on such 28 life stages of Slobozhanshchina medical scientists and naturalists in which they were active in order 29 to shape healthy lifestyle as a value orientation of students and ordinary citizens. The territorial 30 boundaries of the study cover Slobozhanshchina, a historical and geographical region centered in 31 Kharkiv. 32

It was noted that founders of Pharmaceutical education in Ukraine had initiated enlightenment ac-33 tivity for formation of a healthy lifestyle in the educational space and outside of the Imperial Khar-34 kov University. Enlightenment activity of Slobozhanshchyna pharmaceutical scientists in the case 35 of a healthy lifestyle formation was driven by the organization of the Kharkov Medical Society, 36 Kharkov Pharmaceutical Society. 37

It has been determined that the main organizational and pedagogical conditions for increasing the 38 role of scientists of a modern institution of Pharmaceutical education in the formation of a healthy 39 lifestyle should include: the development of readiness of teachers of an institution of higher phar-40 maceutical education to form a healthy lifestyle in the University space and beyond; the introduc-41 tion of health-saving educational technologies in the University space; the study of the health status, 42 promoting the prevention of diseases of students and teachers. 43

Taking into account the problem under study such innovative forms of training as lecture-visuali-44 zation, lecture-excursion; “master classes” and scientific picnics as varieties of independent work ofstudents, participation of students and teachers in the work of international summer camps, etc., 46 have turned out to be effective in the University. 47

It has been proposed to define the concept of “orientation to a healthy lifestyle of representatives of 48 the institution of higher pharmaceutical education”. It includes the selective attitude of employees 49 and students to a healthy lifestyle as a set of material, social and spiritual benefits and ideals con-50 sidered as objects of purpose (implementation of the mission of the University) and tools to meet 51 the needs of both University representatives and ordinary citizens. The attention is drawn to the 52 fact that the strategic task of the National University of Pharmacy (NUPh) is to create a student-53 centered space at the University. The educational activity of the NUPh scientists is aimed at training 54 future pharmacy specialists as professionals and socially responsible citizens, conducting research 55 in accordance with the main global trends in the development of healthcare; providing expert and 56 advisory services to civil society and the authorities. An important direction in implementing the 57 mission, vision and strategic goal of the NUPh is to involve scientists in achieving social well-being, 58 socio-economic development of the region, and educating the society through the formation of a 59 healthy lifestyle as a social orientation. 60

A survey of pharmacists-interns at the premises of the Institute for Advanced Training of Pharmacy 61 Specialists (IATPS) of the NUPh using a questionnaire has revealed the presence of indicators indi-62 cating the respondents’ awareness of the value orientation to a healthy lifestyle. Among these indi-63 cators there is awareness; progress; positivity; manageability; sense of purpose. 

Comment: The abstract is too lengthy and not well written-the focus of the paper is not highlighted. It still have the same structure like the previously returned article.

  1. Materials and Methods

 The historical research methods (chronological, logico-historical, retrospective) al-91 lowed to substantiate the territorial, chronological boundaries of the study, provided con-92 sideration of the subject of research in dynamics. The territorial boundaries of the study 93 cover Slobozhanshchyna – a historical and geographical region centered in Kharkiv 94 (Kharkov on the Imperial era). The chronological boundaries of the study determined by 95 the origin of traditions of pharmaceutical scientists concerning formation a healthy life-96 style in Slobozhanshchyna during 19th – early 20th cent. Attention to the enlightenment 97 activity of modern Slobozhanshchyna pharmaceutical scientists is determined by the fact 98 that on the present period, when the restructuring of higher Pharmaceutical education in 99 Ukraine is taking place, traditions of the socially useful activity of scientists-pharmacists 100 are reviving. 101

General scientific research methods (analysis, synthesis, comparison, classification, 102 systematization, generalization of legal documents, scientific publications, historical and 103 reference publications) were used throughout the study. They allowed to characterize the 104 basic concepts of research. 105

The retrospective method of research allowed to focus on those approaches of the 106 founders of Pharmaceutical education in Ukraine to the problem of formation of a healthy 107 lifestyle as a value orientation in historical retrospect that have special scientific and prac-108 tical value for the present. 109

The personalized method of research has led to a careful study of biographies of 110 Slobozhanshchina medical scientists and naturalists. This method allowed to concentrate 111 attention on such life stages of scientists in which they were active in order to shape 112 healthy lifestyle as a value orientation of students and ordinary citizens. 113

The retrospective and personalized methods of research have been utilized through 114 research sources: 115

− speeches, reports, historical essay of scientists (L. Vannoty, G. Lagermark) [15: 16; 17]; 116

− historical and reference publications about the history of Kharkov University`s 117 Medical Faculty in the first hundred years of its existence (1805–1905) [18], 118

− documents of the regional public and enlightenment societies (19th – early 20 th centu-119 ries), especially of the Kharkov Medical Society, Kharkov Pharmaceutical Society, in the peri-120 odicals of the Imperial era («Харьковский медицинский журнал» (“Harkovskiy med-121 itsinskiy zhurnal”) [19]; newspapers «Южный край» (“Yuzhnyj kraj”) [20], «Харьковские 122 губернские ведомости» (“Harkovskie gubernskie vedomosti”) [21]; 123

− data on the exhibition and museum activity of medical scientists and naturalists of the 124 Imperial Kharkov University [22; 23]; 125

− visual sources (photos, certificate from private collections of NUPh pharmaceutical sci-126 entists); 127

− interpretive and research sources: thesis, articles, monographs of leading scientists re-128 lated to the research topic [10; 12; 18; 24]. 129

The empirical research method (questioning and analytical processing of ques-130 tionnaires) was used for questioning of pharmacists-interns on the problem of forming a 131 healthy lifestyle involving researchers and the teaching staff of the NUPh, as well as ana-132 lytical processing of questionnaires. In the period from June to October 2021, we con-133 ducted a study based on a questionnaire developed by the authors of the article among 134 pharmacists-interns of the NUPh (Kharkiv). The purpose of the survey was: 1) the analysis 135 of the peculiarities of the formation of a healthy lifestyle of the students – future specialists 136 in the Pharmaceutical industry; 2) giving recommendations for improving the effective-137 ness of health-saving measures in the University educational environment, 3) substantia-138 tion of the role of the academic staff in involving pharmacy professionals to change the 139 behavior of ordinary citizens in relation to their own health. 140

It should be noted that during the survey period, pharmacists-interns already had a 141 higher pharmaceutical education. The questionnaire was sent out using the Google form 142

MAJOR COMMENT

The empirical research method (questioning and analytical processing of ques-130 tionnaires : A survey of pharmacists-interns at the premises of the IATPS NUPh using a questionnaire has revealed the presence of indicators showing the respondents’ awareness of the value orientation to a healthy lifestyle. Among these indicators there are awareness; progress; positivity; manageability; sense of purpose. It has been found that the example of well-known scientists-pharmacists and teachers of the students – future specialists in Pharmacy is an important factor in the formation of a healthy lifestyle of Pharmaceutical education applicants.

Suggest to present survey study as a standalone paper.

 The historical research methods (chronological, logico-historical, retrospective) , The retrospective method , General scientific research methods and the personalized method are non-research review that can be a review paper by itself.

Table 1. Criteria and explanation of the questionnaire work : suggest to do away with the table. The info should be described in text.

Round 2

Reviewer 2 Report

All comments have been satisfactorily addressed 

This manuscript is a resubmission of an earlier submission. The following is a list of the peer review reports and author responses from that submission.

Round 1

Reviewer 1 Report

This article is a narrative papers on the education programme conducted in Ukraine.

I have some major issues with this report

Abstract

The article focuses on the role of scientists in the institution of higher pharmaceutical education in achieving the social well-being, in particular in the formation of a healthy lifestyle as a value orientation. The following set of research methods was used: general scientific; historical and pedagogical, personalized, prognostic; empirical (questioning of pharmacists-interns and analytical processing of questionnaires). The historical and pedagogical method allowed to focus on those approaches of the founders of Pharmaceutical education in Ukraine to the problem of formation of a healthy lifestyle as a value orientation that have special scientific and practical value for the present. The territorial boundaries of the study cover Slobozhanshchina, a historical and geographical region centered in Kharkiv. The chronological boundaries of the study cover the 19th – early 21th centuries.

It has been determined that the main organizational and pedagogical conditions for increasing the role of scientists of a modern institution of Pharmaceutical education in the formation of a healthy lifestyle should include: the development of readiness of teachers of an institution of higher pharmaceutical education to form a healthy lifestyle in the University space and beyond; the introduction of health-saving educational technologies in the University space; the study of the health status, promoting the prevention of diseases of students and teachers.

Taking into account the problem under study such innovative forms of training as lecture-visualization, lecture-excursion; “master classes” and scientific picnics as varieties of independent work of students, participation of students and teachers in the work of international summer camps, etc., have turned out to be effective in the University.

It has been proposed to define the concept of “orientation to a healthy lifestyle of representatives of the institution of higher pharmaceutical education”. It includes the selective attitude of employees and students to a healthy lifestyle as a set of material, social and spiritual benefits and ideals considered as objects of purpose (implementation of the mission of the University) and tools to meet the needs of both university representatives and ordinary citizens. The attention is drawn to the fact that the strategic task of the National University of Pharmacy (NUPh) is to create a student-centered space at the University. The educational activity of the NUPh scientists is aimed at training future pharmacy specialists as professionals and socially responsible citizens, conducting research in accordance with the main global trends in the development of healthcare; providing expert and advisory services to civil society and the authorities. An important direction in implementing the mission, vision and strategic goal of the NUPh is to involve scientists in achieving social well-being, socio-economic development of the region, and educating the society through the formation of a healthy lifestyle as a social orientation. A survey of pharmacists-interns at the premises of the Institute for Advanced Training of Pharmacy Specialists (IATPS) of the NUPh using a questionnaire has revealed the presence of indicators indicating the respondents’ awareness of the value orientation to a healthy lifestyle. Among these indicators there is awareness; progress; positivity; manageability; sense of purpose.

Comment: The abstract is not well written-the focus of the paper is not highlighted

  1. Materials and Methods

Comment: This part consists a mixture of Results and Description of the study/background. It spans over 12 pages

  1. Results

A survey of pharmacists-interns at the premises of the IATPS NUPh using a questionnaire has revealed the presence of indicators showing the respondents’ awareness of the value orientation to a healthy lifestyle. Among these indicators there are awareness; progress; positivity; manageability; sense of purpose. It has been found that the example of well-known scientists-pharmacists and teachers of the students – future specialists in Pharmacy is an important factor in the formation of a healthy lifestyle of Pharmaceutical education applicants.

Comment:The main results should be reported here

  1. Discussion

It has been proposed to define the concept of “orientation to a healthy lifestyle of

representatives of the institution of higher Pharmaceutical education”. It includes the selective attitude of employees and students to a healthy lifestyle as a set of material, social and spiritual benefits and ideals considered as objects of purpose (implementation of the mission of the University) and tools to meet the needs of both University representatives and ordinary citizens.

Comment:The discussion section of the results should be reported he

Reviewer 2 Report

Wellbeing is such an important topic.  This paper contributes by outlining the role of scientists in helping to build healthy habits among students and the public.  I appreciated the definitions for value, values and value orientation. The text also provides good context for the study, explaining the school and its goals.

Structure and Labelling:  The paper uses a variety of methods.  Please use a heading to denote each method.  I found myself continuing to look back at the methods list and attempt to see where we were in the methods that had been named.  It seems that pedagogical comes before historical in the text, but they are presented in reverse order in the methods listing.  Again, clustering each type of method together, labeling it and ordering according to the listing you gave, will be helpful to the reader.

Explain Methods:  The methods are named (lines 82-92), but the methods are not described.  How was the historical work conducted? (Who did it, what resources, analytical process, etc.)  Likewise for each method.  The reader needs to know how you came to this information you are presenting.  In addition, I did not see the prognostic and personalized methods described or articulated in the findings.  Each method named should have a corresponding section to the paper where it is elaborated upon.

Historical Section: While this section is very interesting and rewarding for other scientists to see, it is quite long.  Consider how it might be made more condensed.  For instance, shorter summaries of the individuals could be consolidated in a table.  It may also be helpful to consider a figure that is a timeline of events to further help the reader.  Please end this section with statements on how history has informed the present.  As currently written, the transition is a bit abrupt.  But, I’m sure a few statements of how this groundwork influenced the current activities could be made.

Questionnaire:  The historical and document based methods can obviously be conducted without ethics review or consent.  However, it is not clear whether the questionnaire work had appropriate oversight and vetting by officials.  Please explain.  Typical survey reporting guidelines are not followed here.  For instance, what was the response rate?  We were only given the number of respondents.  Was the survey piloted for clarity?  How did you reduce the possibility of duplicate submissions from the same individual?  How were the data cleaned and managed?  What was done with missing responses?  Here is a good resource to help with the reporting:  https://www.ncbi.nlm.nih.gov/pmc/articles/PMC1550605/.  If you are not able to clarify the rigor of the methods, the paper could be considered without this section.  Some questions about the reporting of findings are below. 

Discussion:  The discussion section is very short and does not reflect the scope of your work.  The aim was to analyze the role of scientists.  What has been learned when you examine across the various parts of your study?  What opportunities does this open to further investigation?  What are the limitations of your work?

General flow:  The paper seems to include the past, present and a questionnaire.  It would be helpful to consider more how these pieces fit together (e.g. Discussion) and the implications of the three parts of work.  Transitions between the sections (one identified above in historical comment) would also be helpful.

Some additional points:

- The focus on teacher development is particularly helpful.  Within this topic, I did not understand “frontal forms”.  Please describe differently/further. 

-Lines 167-183 appear to be a long list of programs.  However, the punctuation and length makes it difficult to understand.  Consider another method of presentation such as bullet pointed list or table.

-The concept of a “health-saving” educational environment is interesting and merits further description.  What does this entail?

-Line 342 seems to being a new section, commenting on the current activities with the public.  Please add a suitable heading here to help the reader understand the transition from the historical to present.

-Within the current activities section, it would be helpful to name the activity under discussion at the beginning of each paragraph.  This could be within the first sentence or as a subhead.

-Line 382 seems to start a new section again, commenting on the training of pharmacists.  Please add a suitable heading here to help the reader understand the transition from public to student training.

-Line 424 begins the discussion of post-graduate training.  It would be helpful, earlier in the paper, to describe the educational system for pharmacists.  How many years of pre-pharmacy training, pharmacy, internship? Etc.  This will help the reader appreciate any similarities or differences with their own system. This addition doesn’t need to be long, but would give the general structure.

-Line 455.  It appears that the results begin here, not line 522.

-Line 481.  Was this a question within the questionnaire?  If you are reporting findings from a question, please provide the numerical values related to each item in your list e.g. museums.  If it was a content analysis of qualitative comments, please describe your process.

-Line 496.  How were these indicators identified?  What was the data source and process for conducting the analysis that resulted in this list?

Thank you for the opportunity to review this ambitious work.  You have used diverse approached to understanding the role of scientists!